# Comparison among U-17, U-20, and Professional Female Soccer in the GPS Profiles during Brazilian Championships

**DOI:** 10.3390/ijerph192416642

**Published:** 2022-12-11

**Authors:** Ronaldo Kobal, Leonardo Carvalho, Raíssa Jacob, Marcelo Rossetti, Lucas de Paula Oliveira, Everton Crivoi Do Carmo, Renato Barroso

**Affiliations:** 1School of Physical Education, University of Campinas, Campinas 13083-851, Brazil; 2Sport Club Corinthians Paulista, São Paulo 03087-000, Brazil; 3School of Physical Education and Sport of Ribeirão Preto, University of São Paulo, Ribeirão Preto 14040-900, Brazil; 4Department of Physical Education, Senac University Center, São Paulo 04696-000, Brazil

**Keywords:** football, team sports, younger female players, external load, global positioning systems

## Abstract

The purpose of this study was to compare and characterize the physical demand of official matches among under-17 (U-17), under-20 (U-20), and professional (Pro) female soccer players. All matches were from the U-17, U-20, and Pro National Brazilian Championships. Fourteen Pro matches, nine U-20 matches, and four U-17 matches were analyzed. The external load was measured by the global positioning system (GPS) and the internal workload was assessed by the ratings of perceived exertion (RPE) multiplied by the duration of the match. The activity profiles measured were total distance covered (km), total sprint distance (m) (speed > 18 km·h^−1^), number of accelerations and decelerations (between 1 and 2 m·s^−2^ and >3 m·s^−2^), and top speed (km·h^−1^). For the analysis, we standardized all the metrics (except the top speed) by the time (in minutes) played. The Pro group presented higher sprint distances, number of accelerations and decelerations, and top speeds, compared to U-20 and U-17. There was no difference in the total distance among groups, and there was no difference in any GPS metrics between U-20 and U-17. The RPE was higher in Pro and U-17, compared to U-20; however, the workload-RPE was higher in Pro, compared to both U-17 and U-20 groups. These findings provide important information for the evolution of physical performance according to age categories in elite female soccer players.

## 1. Introduction

Soccer is one of the most popular sports in the world. Two hundred and eleven countries are affiliated to FIFA (Fédération Internationale de Football Association). It is estimated that there are over 300,000 clubs and 240 million registered players worldwide. Recently, a growing number of women playing soccer has been observed. In 2020, FIFA created a development program for female’s soccer that emphasized the importance of fostering young female soccer players and aims at having 60 million women and girls playing soccer by 2026. These numbers highlight the importance of investigating female soccer players to improve their performance.

Soccer is an intermittent team sport with alternating moments of low- and high-intensity activities. For soccer players to perform at their best, they need an optimal level of aerobic and anaerobic fitness, such as endurance, strength, power, and speed abilities [1,2,3]. Some studies reported that during a soccer match, elite women players cover ~10 km, with around ~1500 m of this at high-intensity running speed (i.e., 13–18 km·h^−1^) and ~200 m at sprinting speed (i.e., >18 km·h^−1^) [4]. These findings provide appropriate information for strength and conditioning staff to plan training practices for elite teams.

Global positioning systems (GPS) have been extensively used to measure the external load during soccer matches and to characterize the profile of soccer [5,6]. However, data that compare the physical demands of different age groups of female soccer players are still scarce. Vescovi [7] compared the physical demands of matches in youth female soccer (i.e., U-15, U-16, and U-17) and reported that U-15 players covered significantly lower total and sprint distances than U-16 and U-17 players (i.e., 6.96 < 8.02 and 8.55 km; 76 < 185 and 235 m). Although this study provides benchmarks for young female athletes, it does not present a comparison of different age groups toward the elite level.

We are aware of only one study that compared match activity profiles in U-17, U-20, and professional female soccer players [8]. These data were obtained during official international competitions of the Brazilian national teams. Overall, the physical demands were different between age groups (Professional > U-20 > U-17). These findings suggest match-specific characteristics according to age group. Nevertheless, players from national teams follow training programs from distinct clubs, and these data may not provide the specific development of a long-term training program for young female players. Thus, it is important to identify characteristics from soccer matches from a national-level club that has a female soccer development program, so it can provide some insights into the training process.

In addition, identifying the profile of external (e.g., GPS metrics) and internal (e.g., RPE) load across age groups could help plan the training and the recovery process and minimize fatigue across matches [9]. For instance, Kobal, Aquino, and Carvalho et al. [9] investigated the effect of the number of substitutions during official matches on internal load and recovery in elite female soccer players. They observed that with a higher number of substitutions, workload (estimated by the session-RPE method) was lower, which may have contributed to the improved recovery status of the players. Thus, examining the response of the internal load to the physical demands presented by official matches (i.e., external load) of young and professional players presents additional information to characterize the developmental process of female soccer players and may provide benchmarks for other teams.

Therefore, the aims of this study were twofold: to characterize the activity demands of elite female soccer players and to compare the GPS motion profiles and internal load (i.e., measured by RPE) among U-17, U-20, and professional female soccer players in the Brazilian age group championship. Moreover, the present study examined the associations between workload and all GPS measures. According to previous studies, it was hypothesized that professional players would present higher results from GPS metrics than younger players, with minimal differences between U-17 and U-20.

## 2. Materials and Methods

### 2.1. Study Design

This retrospective study was conducted in the 2022 season during the National Brazilian Championships of U-17, U-20, and professional (Pro) female soccer players. Fourteen matches from Pro, nine matches from U-20, and four matches from U-17 were analyzed. We used GPS data from all the players that participated in each match. Due to the different match durations (90 min for Pro, 80 min for U-20, and 70 min for U-17), we standardized the GPS metrics by the match duration, except for the top speed.

### 2.2. Participants

Twenty-three professional (28.0 ± 4.6 years, 165.3 ± 5.2 cm, 59.1 ± 5.4 kg, and body fat 14.8 ± 2.0%), twenty-one U-20 (17.8 ± 0.7 years, 161 ± 8.0 cm, 61.6 ± 8.2 kg, and body fat 20.4 ± 3.0%), and twenty-four U-17 (16.1 ± 0.7 years, 162 ± 6.3 cm, 55.8 ± 5.5 kg, and 18.0 ± 3.4%) from the same club participated in this study. All players participated in the National Brazilian Championships of their respective age group during the 2022 season. Goalkeepers were excluded from the study. All participants were informed about the procedures, benefits, discomforts, and possible risks of the study and signed a from signaling their free and informed consent before participation. The University’s Research Ethics Committee approved the experimental protocol.

### 2.3. Rating of Perceived Exertion (RPE)

Athletes were familiarized with the CR-10 Borg RPE scale as this was part of their training routine. RPE was assessed with the modified Borg 10-point (0–10) scale, which is widely used in both practical and research settings [10]. This method uses a simple question: “How was your match today?”. The answer was provided 30 min after the end of training sessions and official matches, by choosing a descriptor and a number from 0–10. We considered the mean of each match in the analysis. The workload-RPE was determined by multiplying each players’ playing time (minutes) in each match by the RPE, as described by Foster, Florhaug, Franklin et al. [10].

### 2.4. Match Running Performance

All soccer matches activity profiles were obtained via a GPS system operating at 10 Hz (GPS units; Playertek, Catapult Innovations, Melbourne, Australia). GPS devices were fitted to the upper back of players using a harness and the same unit was used by each player in all measures to reduce inter-unit measurement error. Units were turned on 10 min before each match. The activity profiles measured were total distance covered (km), total sprint distance (m) (speed > 18 km·h^−1^), number of accelerations and decelerations (between 1 and 2 m·s^−2^, and >3 m·s^−2^), and top-speed (km·h^−1^). For the analysis, we standardized all the metrics, except the top speed, by the time played; total distance and sprint distance are presented as m·min^−1^, number of accelerations·min^−1^, number of decelerations·min^−1^.

### 2.5. Statistical Analysis

Data were visually inspected for the existence of outliers (i.e., box plots), tested for normality (Shapiro–Wilk) and homogeneity (Levene), and are presented as mean standard deviation (SD). All data of perceived exertion and physical parameters for the 3 conditions, professional (Pro), under-20 (U-20), and under-17 (U-17), were analyzed with one-way repeated measures ANOVA. When a significant F value was found, Tukey post hoc was used for multiple comparisons. Additionally, effect size (ES) was calculated to compare variables across age groups according to Cohen [11] and classified according to Rhea [12]: trivial (<0.25), small (0.25 to 0.50), moderate (>0.50 to 1.0), and large (>1.0). A Pearson product–moment test was performed to determine the relationships among the workload and all GPS metrics in each age group. The correlation coefficients were qualitatively interpreted, when significant, as follows: <0.09, trivial; 0.10–0.29, small; 0.30–0.49, moderate; 0.50–0.69, large; 0.70–0.89, very large; >0.9 nearly perfect. The level of significance was set at *p* < 0.05. Statistical analyses were performed using the software package IBM SPSS (V. 25, SPSS Inc., Chicago, IL, USA).

## 3. Results

The total distance·min^−1^ and the total sprint distance·min^−1^ covered are presented in Figure 1. There was no difference in total distance·min^−1^ among age groups (F = 1.875; *p* = 0.157). The total distance covered by Pro (95.9 ± 13.9 m·min^−1^), U-20 (94.7 ± 11.6 m·min^−1^), and U-17 (97.0 ± 23.2 m·min^−1^) can be observed in Figure 1A. Among the group, a difference was observed in the total sprint distance·min^−1^ (F = 23.19; *p* < 0.0001; Figure 1B), which was higher in Pro (7.8 ± 2.8 m·min^−1^), compared to U-20 (6.3 ± 2.6 m·min^−1^; *p* < 0.001; *d* = 0.59) and U-17 (5.5 ± 2.5 m·min^−1^; *p* < 0.001; *d* = 0.81).

Table 1 displays values of top speed, number of accelerations·min^−1^, and number of decelerations·min^−1^. Top speed was different between groups (F = 13.24; *p* < 0.0001) with higher values observed in Pro than in U-20 (ES = 0.53, moderate) and U-17 (ES = 0.49, small). There was no difference between U-20 and U-17. There was no difference in the number of accelerations.min^−1^ when considering acceleration between 1 and 2 m·s^−2^ (F = 1.558; *p* = 0.211). However, when accelerations > 3 m·s^−2^ were analyzed, higher values were observed in Pro, compared to U-20 (ES = 1.06, large) and U-17 (ES = 0.88, moderate) (F = 52.68; *p* < 0.0001), without significant differences between U-20 and U-17. We observed between-group differences in the number of decelerations·min^−1^ in both 1–2 m·s^−2^ (F = 5.819; *p* = 0.0032), and >3 m·s^−2^ (F = 27.87; *p* < 0.0001). The number of decelerations.min^−1^ was higher in Pro and U-17, compared to U-20 for 1–2 m·s^−2^ (0.40, small) and Pro was higher for >3 m·s^−2^ (0.76, moderate) compared to U-20 and U-17 (0.73, moderate), without difference between U-20 and U-17.

RPE was different among groups, Pro and U-17 (*p* < 0.001) were higher compared to U-20 (Figure 2A). Workload-RPE was different among age groups (F = 23.04; *p* < 0.0001; Figure 2B). The workload-RPE was higher in Pro, compared to U-20 (d = 0.82, moderate, *p* < 0.05) and U-17 (d = 0.75, moderate, *p* < 0.05). There was no difference between U-20 and U-17 (*p* = 0.11).

Table 2 shows correlations coefficients between the workload-RPE and the GPS metrics [total distance covered (m·min^−1^), total sprint distance (m·min^−1^), top speed (km·h^−1^), and number of accelerations and decelerations between 1 and 2 m·s^−2^ and >3 m·s^−2^ (n·min^−1^)]. Trivial-to-moderate differences were obtained between workload and all GPS metrics.

## 4. Discussion

This study compared the match activity analyzed by GPS and the workload in three age groups (Pro, U-20, and U-17) of elite female soccer players during official soccer matches. Our main findings were: (a) there was no difference in total distance covered among age groups; (b) Pro covered higher sprint distance.min^−1^ and higher number of accelerations and decelerations.min^−1^ and achieved faster top speed than U-20 and U-17; (c) workload-RPE was higher in Pro than U-20 and U-17; (d) no differences were observed between U-20 and U-17 in the GPS metrics; (e) trivial-to-moderate association was found between the workload and all GPS metrics in each age group.

The total distance covered by players was not different among age groups. These findings differ from those of Ramos, Nakamura, Penna et al. [8]. These authors investigated match activity of Pro, U-20 and U-17 Brazilian National women teams during international championships and reported that Pro (i.e., Rio 2016 Olympic games) covered a larger total distance than U-20 and U-17 (both South American championships). In their study, all matches lasted 90 min for all age groups. In our study, we chose to standardize total distance by the minutes played, because of the differences in the duration of the matches of each age group (2 × 45 min for Pro, 2 × 40 min for U-20, and 2 × 35 min for U-17). A possible explanation for the lack of difference in the total distance covered might lie in the distinct duration of the matches. Some studies have shown that the physical performance of players decline as the matches progress, especially at the end of the second half [3,13,14]. As the match duration is shorter for younger players, they are likely to spend less time in the critical period of the game, which can prevent impairment in their physical performance, and, as a result, when standardized by the match duration, we did not observe any differences.

Conversely, when we analyzed all data that are related with high-intensity activities (top speed, sprint distance, and the number of accelerations and decelerations), we observed that these metrics were higher in the Pro group, compared to U-20 and U-17 groups. These results show that the physical demands of the most important actions in games are superior in the Pro group [15]. This difference between Pro and younger players may be a consequence of the biological maturation and/or development of motor abilities throughout the aging process [16,17,18]. Taken together, these findings suggest that muscle strength and power capabilities play an important role in soccer performance, since experienced players can better execute power-related motor tasks than younger players.

For instance, more mature young soccer players perform a larger number of high-intensity actions and achieve faster maximal sprint speed, compared to their less mature peers [8,16,19]. It is possible that the higher high-intensity game metrics observed are positively associated, at least partly, with the maturational status. Although tempting, we can only speculate as we did not assess the maturation stages of our participants. These data demonstrate that each age group seems to present specific needs, and strength and conditioning staff should be aware of them to design the best training program.

Strength and power production is improved by long-term training, and increases in this abilities induce positive changes in speed, acceleration, and deceleration [20]. Ramos, Nakamura, Penna, Mendes, Mahseredjian, Lima, Garcia, Prado, and Coimbra [16] investigated the power capabilities tests (linear sprint, vertical jump, and change of direction) in women soccer players of different ages (Pro, U-20, U-17, and U-15) and revealed that older age groups were faster in both linear sprint and change of direction and they jumped higher, compared to younger groups (Pro > U-20 > U-17 > U-15). It should be noticed that we investigated the number of accelerations and decelerations of different magnitudes (between 1 and 2 m·s^−2^, and >3 m·s^−2^). When the lower-magnitude accelerations and decelerations (1–2 m·s^−2^) are compared to higher-magnitude (>3 m·s^−2^) accelerations and decelerations, some interesting findings emerge, as there was no between-group difference in acc1–2 m·s^−2^ and dec1–2 m·s^−2^, but the number of accelerations and decelerations > 3 m·s^−2^ were higher in Pro. Considering that the level of strength/power is associated with the speed ability [21], this result supports a possible influence of strength/power levels in the different high-intensity physical demands of soccer games, especially in terms of top speed, acceleration, and deceleration.

In addition, the metrics of high-intensity actions may also be affected by aerobic endurance [2,22,23]. Krustrup, Mohr, Amstrup et al. [22] reported correlations between the Yo-Yo test and the number of high-intensity actions (r = 0.71), number of sprints (r = 0.58), and total distance covered (r = 0.53) during a soccer match. Further, professional women soccer players covered longer distances in the Yo-Yo intermittent test, compared to U-20, U-17, and U-15 players [16]. Thus, the higher demands in high-intensity metrics observed herein may reflect higher aerobic endurance of professional players. One possible explanation is that Pro players possess greater ability to support high-intensity actions from the long-term physiological adaptations provided by specific soccer training, which allows a higher frequency of occurrence of these actions throughout the match and increases the total demand for these metrics at the end of the game. Therefore, it is possible that the cumulative effect of long-term training and exposure to the game demands induced increases in high-intensity metrics and generates between-group differences.

Furthermore, the RPE after the matches indicated that both Pro and U-17 present higher RPE, compared to U-20. As RPE is affected both by intensity and duration of a task [24], the shorter duration of the U-20 matches, compared to Pro, could explain the differences and also suggest that the length of the match of the U-20 could be increased to similar values of the Pro games to elicit similar RPE values. However, the differences in match duration does not explain the difference when compared to U-17 (lower RPE in U-20 players). We suggest that the similar relative GPS metrics between these age groups (U-20 and U-17) are more stressful in younger female players, possibly due to some age-related explanations. Alternatively, as the CR-10 scale was developed to reflect both physiological (oxygen uptake, ventilation, HR, circulating glucose concentration, and glycogen depletion) and psychological responses to exercise [25], we propose that playing an official match at the highest national level imposes a psychological stress in the younger players, which is expressed at higher RPE. These findings suggest aging may affect not only physical performance but also players’ psychological characteristics.

Further, our results also reveal that Pro presented higher workload-RPE (i.e., RPE × time played), compared to both U-20 and U-17. Even though the duration of the match was longer for Pro, which affects the estimation of workload-RPE, it is possible that this latter finding may be a consequence of the higher physical demands of the game observed in Pro. The workload-RPE indicates that the internal load, which represents the physiological stress imposed on the athlete in response to the exercise stimulus, is positively related to the external load in sports modalities, such as running and swimming [26]. However, similar findings have not been observed in soccer [27,28], where the low or non-existent associations between internal load and top speed, sprint distance, and numbers of acceleration were low or non-existent [27,28]. In fact, our findings show only few significant correlations between GPS metrics and workload, suggesting that metrics alone do not provide a good indication of the physiological stress imposed on players. Although it was not within the scope of this investigation, external and internal loads may be related to the occurrence of injury, as higher loads may be related to greater risks of incidence of injury [29]. Thus, we suggest that future studies explore the possible relationship between loads and injuries in women soccer players.

This study is not without limitation. We only investigated one club in the Brazilian League as there are not many U-17 and U-20 teams, and even fewer have GPS units available for the younger players. Further studies are encouraged to investigate the match-characteristics of different age groups in different clubs. Our analysis focused only on the match characteristics; it would be highly interesting to investigate and compare data from training sessions in different age groups.

## 5. Conclusions

This study suggest that professional female soccer players showed higher physical performance during official matches, especially in metrics related to the high intensity demand of the game, such as sprint distance, top speed, acceleration, and deceleration, compared to U-20 and U-17. In addition, the RPE and workload-RPE demonstrated specific responses by the group that became an important tool for measuring the internal load in all age groups. These results have important practical implications and can be used by technical staff of women’s soccer teams as the characterize profile of external load demands obtained in the matches of professional, U-20, and U-17, thus monitoring the evolution of players within the long-term training process can determine the balance between the external and internal load of continuum processes to the development of female soccer players.

## Figures and Tables

**Figure 1 ijerph-19-16642-f001:**
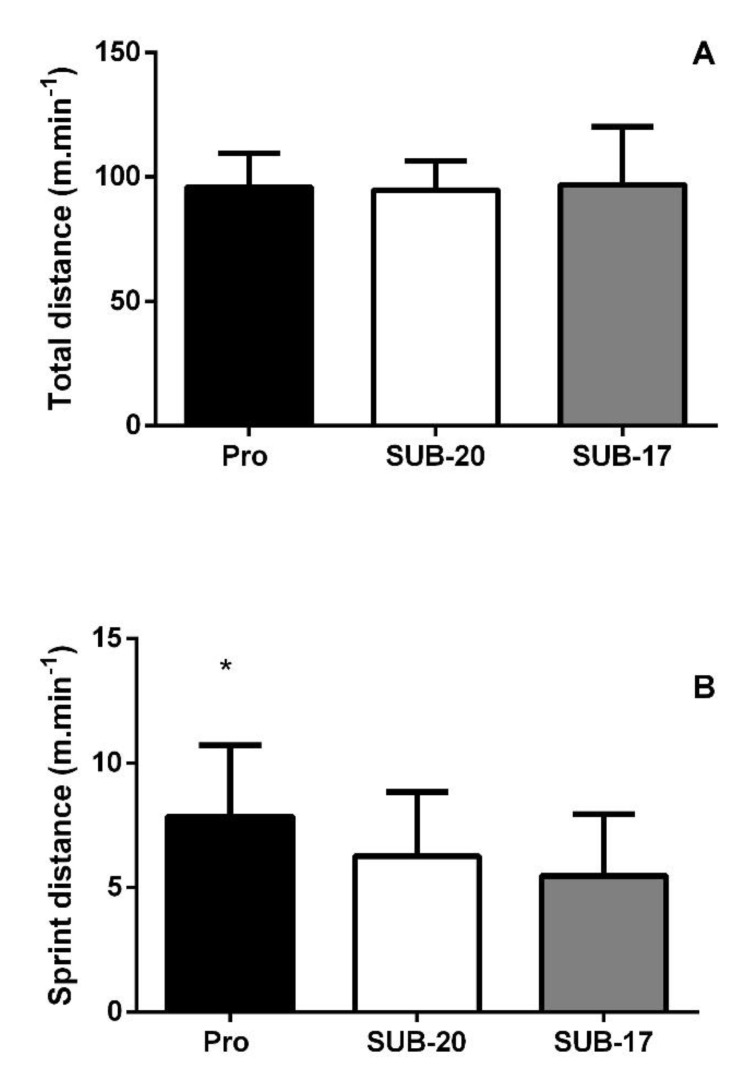
(**A**) Total distance (m·min^−1^) and (**B**) Sprint distance (m·min^−1^) in Pro, under-20 (U-20), and under-17 (U-17) groups. * *p* < 0.05 vs. U-20 and U-17.

**Figure 2 ijerph-19-16642-f002:**
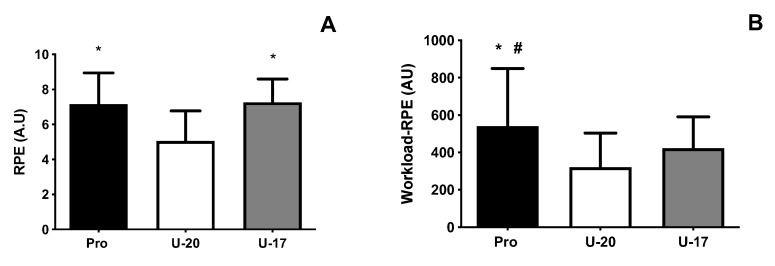
(**A**) RPE and (**B**) Workload-RPE in Pro, under-20 (U-20), and under-17 (U-17) age groups. AU—arbitrary units; * *p* < 0.05 vs. U-20; # *p* < 0.05 vs. U-17.

**Table 1 ijerph-19-16642-t001:** Top speed and number of accelerations and decelerations for three age groups. Values are presented as mean ± sd.

	Pro	U-20	U-17	Pro vs. U-20	Pro vs. U-17	U-20 vs. U-17
**Top Speed (km·h^−1^)**	26.7 ± 2.5 *^, #^	25.5 ± 1.8	25.5 ± 2.1	<0.0001	0.002	0.99
**Acc 1–2 m·s^−2^ (n·min^−1^)**	2.76 ± 0.58	2.69 ± 0.56	2.85 ± 0.76	0.587	0.64	0.28
**Acc > 3 m·s^−2^ (n·min^−1^**)	0.88 ± 0.27 *^, #^	0.57 ± 0.20	0.61 ± 0.21	<0.0001	<0.0001	0.62
**Dec 1–2 m·s^−2^ (n·min^−1^)**	2.53 ± 0.53 *	2.36 ± 0.54	2.59 ± 0.75 *	0.022	0.803	0.04
**Dec > 3 m·s^−2^ (n·min^−1^)**	1.04 ± 0.26 *^, #^	0.83 ± 0.24	0.82 ± 0.32	<0.0001	<0.0001	0.97

Pro—professional, U-20—under-20, U-17—under-17, Acc—number of accelerations, Dec—number of decelerations, *p*—*p* value, ES—effect size. * *p* < 0.05 vs. U-20 ^#^
*p* < 0.05 vs. U-17.

**Table 2 ijerph-19-16642-t002:** Correlations among workload and all GPS metrics.

	Total Dist. (m·min^−1^)	Sprint Dist. (m·min^−1^)	Top Speed (km·h^−1^)	Acc 1–2 m·s^−2^ (n·min^−1^)	Acc > 3 m·s^−2^ (n·min^−1^)	Dec 1–2 m·s^−2^ (n·min^−1^)	Dec > 3 m·s^−2^ (n·min^−1^)
**Pro (workload)**	0.02	−0.18 * Small	0.15	−0.03	−0.18 * Small	0.04	−0.16
**U-20** **(workload)**	−0.04	0.12	0.44 * Moderate	−0.18	−0.08	−0.21 * Small	−0.04
**U-17** **(workload)**	−0.16	−0.09	0.34 * Moderate	−0.22	−0.33	−0.25	−0.18

Pro—professional; U-20—under-20; U-17—under-17; Acc—number of accelerations, Dec—number of decelerations. *—*p* < 0.05. The correlation coefficients were qualitatively interpreted, when significant, as follows: <0.09, trivial; 0.10–0.29, small; 0.30–0.49, moderate; 0.50–0.69, large; 0.70–0.89, very large; >0.9, nearly perfect.

## Data Availability

Data will be made available upon request to the corresponding author.

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
