# Peer review of "Comparison among U-17, U-20, and Professional Female Soccer in the GPS Profiles during Brazilian Championships"

_ijerph, 2022, doi:10.3390/ijerph192416642_

Round 1
Reviewer 1 Report
Thank you very much for allowing me to review this article. In my opinion, the authors create a gap in the literature that they do not fill through their study. They describe in the introduction "We are aware of only one study that compared match activity profiles in U-17, U-20, and professional female soccer players [8]. These data were obtained during official international competitions of the Brazilian national teams. Overall, the physical demands were different between age-groups (Professional > U-20 > U-17). However, players from National teams follow training programs from distinct clubs, and these data may not provide specific development of a long-term training program for young female players."
I agree, national team competitions are not comparable to the league. But in their article, they only analyze players from the same club. This does not resolve the gap, since a league must be represented by several clubs, which will also have different demands among themselves. For the analysis of a league, I recommend, for example, reading "Epidemiology of Injuries in First Division Spanish Women's Soccer Players. Int. J. Environ. Res. Public Health 2021,18, 3009. https://doi.org/10.3390/ijerph18063009", to get a little idea of ​​having a minimum representation of a soccer league.
Author Response
Thank you very much for allowing me to review this article. In my opinion, the authors create a gap in the literature that they do not fill through their study. They describe in the introduction "We are aware of only one study that compared match activity profiles in U-17, U-20, and professional female soccer players [8]. These data were obtained during official international competitions of the Brazilian national teams. Overall, the physical demands were different between age-groups (Professional > U-20 > U-17). However, players from National teams follow training programs from distinct clubs, and these data may not provide specific development of a long-term training program for young female players."
I agree, national team competitions are not comparable to the league. But in their article, they only analyze players from the same club. This does not resolve the gap, since a league must be represented by several clubs, which will also have different demands among themselves. For the analysis of a league, I recommend, for example, reading "Epidemiology of Injuries in First Division Spanish Women's Soccer Players. Int. J. Environ. Res. Public Health 2021,18, 3009. https://doi.org/10.3390/ijerph18063009", to get a little idea of ​​having a minimum representation of a soccer league.
ANSWER: The reviewer has a good point. We agree that data from one club may not represent data from the league. However, the number of women soccer players is growing, the development of players is important and is also of the goals of FIFA’s plan for the development of women soccer. As data regarding women soccer players’ development is still scarce, we believe this study provides important information for coaches, team and federation managers, and can contribute to the development of the sport. We include some information stating that our data refers to a single team in our national league.
Reviewer 2 Report
TITLE: Comparison among U-17, U-20 and Professional female soccer in the GPS profiles during Brazilian championships
KEYWORDS: Could you add some specific word
INTRODUCTION
It is correct and reads clearly.
RESULTS
In general terms, place the tables and figures after the paragraph that talks about them to facilitate the reading of the results.
You would make a change to the RPE results (lines 168 to 172). Now they are written in text, but I find it more interesting that they appear in the form of a figure or table. By contrast, table 2 of correlations would remove it and explain it in text.
Remove the effect size of table 1 and include it in the text to make it clearer. The table shows the ES of all the differences, however, some are not significant, therefore it would not be necessary to put it.
DISCUSSION
Does the job mentioned on lines 238 to 240 have the same characteristics as yours? yes, some are commented but, for example, the sample is also of women? or the analysed matches are from the same competition? these factors could influence subsequent reasoning.
In some of the subsequent arguments, it might be interesting to talk about maturational processes. Could the differences obtained be due to them? strength, speed, power, are they capacities that have already been fully developed at 17 years of age? could condition their results or, at least, their reasoning.
On the other hand, what is the aerobic capacity of your sample? differ between groups? Could this be an explanation why the Pro group is able to sustain higher intensities? further specify this reasoning.
Finally, the reasoning that appears in lines 289 to 293 seems contradictory since the RPE in U20 was different from that of Pro and U17 but always lower, therefore it could not be associated with the duration of the task. That would be possible if U20's RPE was lower than Pro's but higher than U17's.
Author Response
KEYWORDS: Could you add some specific word.
ANSWER: Thank you for your suggestion, we added “younger female players”, “external load”, and “global positioning systems”.
RESULTS
In general terms, place the tables and figures after the paragraph that talks about them to facilitate the reading of the results.
ANSWER: Thank you for your suggestion, we changed accordingly.
You would make a change to the RPE results (lines 168 to 172). Now they are written in text, but I find it more interesting that they appear in the form of a figure or table. By contrast, table 2 of correlations would remove it and explain it in text.
ANSWER: Thank you for your suggestion, we excluded the RPE values in the text and maintained in the figure 2.
The table 2 we kept all data of correlation in table because we believe that it is easier to the reader understand.
Remove the effect size of table 1 and include it in the text to make it clearer. The table shows the ES of all the differences, however, some are not significant, therefore it would not be necessary to put it.
ANSWER: Thank you for your suggestion, we changed accordingly.
DISCUSSION
Does the job mentioned on lines 238 to 240 have the same characteristics as yours? yes, some are commented but, for example, the sample is also of women? or the analysed matches are from the same competition? these factors could influence subsequent reasoning.
ANSWER: Thank you for your suggestion, we include “women” in the sentence, “These authors investigated match activity of Pro, U-20 and U-17 Brazilian National women teams during international championships…” Moreover, we described the competition that each age group played.
In some of the subsequent arguments, it might be interesting to talk about maturational processes. Could the differences obtained be due to them? strength, speed, power, are they capacities that have already been fully developed at 17 years of age? could condition their results or, at least, their reasoning.
ANSWER: Thank you for your suggestion. We included: This difference between Pro and younger players may be a consequence of the biological maturation and/or development of motor abilities throughout the aging process [16-18]. Taken together, these findings suggest that muscle strength and power capabilities play an important role in soccer performance, since experienced players can better execute power-related motor tasks than younger players.
For instance, more mature young soccer players perform a larger number of high-intensity actions and achieve faster maximal sprint speed compared to their less mature peers [8, 16, 19]. It is possible that the higher high-intensity game metrics observed are positively associated, at least partly, with the maturational status. Although tempting, we can only speculate as we did not assess the maturation stages of our participants. These data demonstrate that each age-group seems to present specific needs, and strength and conditioning staff should be aware of them to design the best training program.
On the other hand, what is the aerobic capacity of your sample? differ between groups? Could this be an explanation why the Pro group is able to sustain higher intensities? further specify this reasoning.
ANSWER: Thank you for your comment, we understand the relevance of the aerobic capacity for soccer players performance. However, our study design proposed to investigate the characterization of official matches from three age categories and to compare them. Thus, we did not use any aerobic test in this study. In the previous version, we have included a brief discussion on the importance of aerobic capacity to the soccer matches.
Pro group is able to sustain higher intensities because of some points: 1) the age (± 28 y vs. >17.8 y) and consequently allow higher long term (many years) of soccer training, strength-power abilities and official matches, which enables to improve the physical fitness throughout the process of formation; 2) Pro group compete around 45 matches by season and younger players compete around 20 matches by season, this difference can be determinant to reach higher physical performance; 3) Pro group contain athletes that there is no influence of maturational status, although we did not control the maturational status of our athletes, it is plausible to assume that there is some difference between Pro and younger groups.
Finally, the reasoning that appears in lines 289 to 293 seems contradictory since the RPE in U20 was different from that of Pro and U17 but always lower, therefore it could not be associated with the duration of the task. That would be possible if U20's RPE was lower than Pro's but higher than U17's.
ANSWER: Thank you for your suggestion, we changed the sentence and rewritten. The paragraph now reads:
Furthermore, the RPE after the matches indicated that both Pro and U-17 present higher RPE compared to U-20. As RPE is affected both by intensity and duration of a task [24], the shorter duration of the U-20 matches compared to Pro could explain the differences and also suggest that the length of the match of the U-20 could be increased to similar values of the Pro games to elicit similar RPE values. However, difference in match duration does not explain the difference when compared to U-17 (lower RPE in U-20 players). We suggest that the similar relative GPS metrics between these age-groups (U-20 and U-17) are more stressful in younger female players, possibly due to some. Alternatively, as the CR-10 scale was developed to reflect both physiological (oxygen uptake, ventilation, HR, circulating glucose concentration, and glycogen depletion) and psychological responses to exercise [25], we propose that playing an official match at the highest national level imposes a psychological stress in the younger players which is expressed at higher RPE. These findings suggest aging may affect not only physical performance, but also players’ psychological characteristics.
Reviewer 3 Report
The authors have developed a well-conducted and well-written retrospective study to compare and characterize the physical demand from official matches among under-17, under-20, and professional female soccer players.
However, I suggest two clarifications or modifications that will in my opinion improve the quality of their manuscript:
1. Please, could you add the study design in the title?
2.Please, between lines 67-68, write well the citation of the work.
3. Please, could you add the name of the University to which the Ethics Committee that endorses the study belongs, as well as the registration number?
4. Please, between lines 116-117, write well the citation of the work.
5.I advise the authors to mention in the "Discussion" section a possible invitation for future research that relates external load and internal load data to the risk or prevalence of injuries, especially anterior cruciate ligament injuries...establishing specific characteristics could help prevent this type of injury or others...I suggest that the authors support this argument by citing the following work: DOI: 10.3390/ijerph182312845
5. It is necessary to add a "Limitations" section, and to comment on several that concern the authors' study.
6. Could you separate the "Conclusions" section into important points?
Author Response
The authors have developed a well-conducted and well-written retrospective study to compare and characterize the physical demand from official matches among under-17, under-20, and professional female soccer players.
However, I suggest two clarifications or modifications that will in my opinion improve the quality of their manuscript:
- Please, could you add the study design in the title?
ANSWER: Thank you for your suggestion. However, we believe that the title would be too long, so we chose not to change. We hope that the reviewer understands our choice.
- Please, between lines 67-68, write well the citation of the work.
ANSWER: Thank you for your observation. We fixed it.
- Please, could you add the name of the University to which the Ethics Committee that endorses the study belongs, as well as the registration number?
ANSWER: Thank you for your observation, we included ate the end of the manuscript the Ethical statement. All subjects and/or their legal guardians gave their informed consent for inclusion before they participated in the study. The study was conducted in accordance with the Declaration of Helsinki, and the protocol was approved by the Ethics Committee of the University of Campinas (5.758.886).
- Please, between lines 116-117, write well the citation of the work.
ANSWER: Thank you for your observation, we fixed it.
- I advise the authors to mention in the "Discussion" section a possible invitation for future research that relates external load and internal load data to the risk or prevalence of injuries, especially anterior cruciate ligament injuries...establishing specific characteristics could help prevent this type of injury or others...I suggest that the authors support this argument by citing the following work: DOI: 10.3390/ijerph182312845.
ANSWER: Thank you for your suggestion, we add this suggestion in the Discussion a section as possible future research.
- It is necessary to add a "Limitations" section, and to comment on several that concern the authors' study.
ANSWER: Thank you for your suggestion. We added “This study is not without limitation. We only investigated one club in the Brazilian League, as there are not many U-17 and U-20 teams, and even fewer have GPS units available for the younger players. Further studies are encouraged to investigate the match-characteristics of different age-groups in different clubs. Our analysis focused only on the match characteristics, it would be highly interesting to investigate and compare data from training sessions in different age-groups.”
- Could you separate the "Conclusions" section into important points?
ANSWER: Thank you for your suggestion. We believe that the format of this journal does not allow this type of split.
Round 2
Reviewer 1 Report
The authors have not satisfied the problem of representation of a club in relation to the league.
Reviewer 3 Report
he authors have substantially improved their manuscript over the previous version, so I recommend their work for publication.
Congratulations